# Who Reasons in the Large Language Models?

**Jie Shao**   **Jianxin Wu**[*]

National Key Laboratory for Novel Software Technology, Nanjing University, China
School of Artificial Intelligence, Nanjing University, China
`shaoj@lamda.nju.edu.cn, wujx2001@nju.edu.cn`

## Abstract

Despite the impressive performance of large language models (LLMs), the process of endowing them with new capabilities—such as mathematical reasoning—remains largely empirical and opaque. A critical open question is whether reasoning abilities stem from the entire model, specific modules, or are merely artifacts of overfitting. In this work, we hypothesize that the reasoning capabilities in well-trained LLMs are primarily attributed to the output projection module (`o_proj`) in the Transformer's multi-head self-attention (MHSA) module. To support this hypothesis, we introduce Stethoscope for Networks (SfN), a suite of diagnostic tools designed to probe and analyze the internal behaviors of LLMs. Using SfN, we provide both circumstantial and empirical evidence suggesting that `o_proj` plays a central role in enabling reasoning, whereas other modules contribute more to fluent dialogue. These findings offer a new perspective on LLM interpretability and open avenues for more targeted training strategies, potentially enabling more efficient and specialized LLMs.

## 1 Introduction

Although large language models (LLMs) [29, 6, 42, 5] have exhibited great success and potential in various aspects, developing new capabilities for LLMs [54, 17, 38, 14] is still a trial and error experimentation process in most cases. For example, one of the most exciting milestones is LLMs that can reason [18, 13, 40], e.g., solving complicated mathematical problems using a reasoning sequence that is agreeable by human experts.

This success, however, is still in the black-box style. Currently, there are two primary approaches to inspiring reasoning capabilities in LLMs. For the most advanced models [13, 52], reinforcement learning method (for example, PPO [37], DPO [30], or GRPO [38]) is commonly adopted to enhance the model's ability to solve complex mathematical or programming problems in a step-by-step manner [49]. A more efficient alternative involves supervised fine-tuning (SFT): by providing the backbone LLM with well-prepared, diverse, and step-by-step reasoning traces—often generated through handcrafted examples or existing reasoning models [55, 25, 13, 52]—the model surprisingly acquires reasoning abilities after training. However, despite the practical success of this method, the underlying mechanism remains largely unexplained. It is still unclear why or how this ability emerges. Several potential explanations may account for this phenomenon:

Case 1  Is it the LLM in its entirety (i.e., the union of all its weights) that leads to this capability, such that this miracle is not explainable?

Case 2  Or, is there certain module(s) in it that should be praised for this success, such that we can advance our understanding of LLMs?

---

[*]Corresponding author.

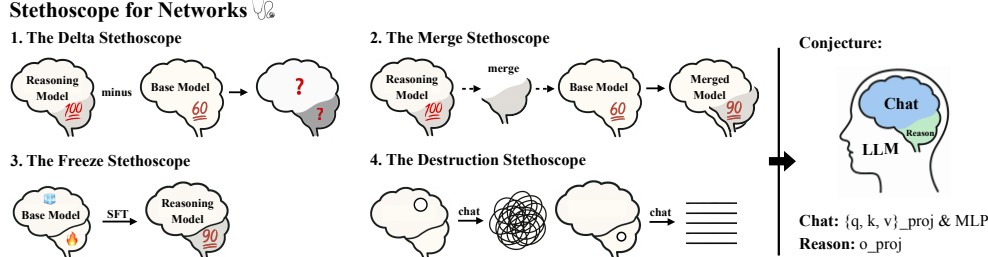

Figure 1: **Stethoscope for Networks.** SfN is a framework designed to identify which components of an LLM give rise to specific abilities. By comparing weight changes and observing behaviors under controlled module merging, tuning, or destruction, SfN provides interpretable insights into the origin of capabilities like reasoning.

> **Case 3** Or in the worst scenario, is reasoning an illusion (e.g., by overfitting to certain types of data), such that we have overestimated the potentials of LLMs?

A definitive answer to any of the above questions will be extremely valuable to guiding the future direction of LLM research. Even a hypothesis or conjecture supported by circumstantial evidences will be highly enlightening, too, let alone when convincing empirical evidences are available.

To this end, our hypothesis is that Case 2 holds in LLMs that reason well. To be more precise, we hypothesize that it is the output projection's parameters (`o_proj`) in the Transformer [44]'s multi-head self-attention (MHSA) module that is in charge of reasoning in an LLM.

To support our hypothesis, we propose a few techniques for diagnosing LLM's behaviors, in particular, the potential functionalities and impacts of various modules in it. We call these techniques Stethoscope for Networks, or SfN (summarized and illustrated in Figure 1). Starting from reasoning-enhanced models, we argue that the weight differences between a base LLM and its fine-tuned counterpart (e.g., for reasoning tasks) provide firsthand and crucial evidence for understanding internal changes. We refer to this approach as the Delta Stethoscope.

In addition, we introduce two novel and previously unexplored methods within the SfN framework: the Merge Stethoscope and the Destruction Stethoscope. The Merge Stethoscope replaces specific modules in a base model with those from a reasoning model. Surprisingly, the resulting variant can maintain fluent dialogue and demonstrate improved reasoning ability in some cases. This phenomenon offers strong clues about the origin and localization of reasoning capability in LLMs. The Destruction Stethoscope, in contrast, systematically disables individual modules and observes the resulting behavior to infer the functional roles of each component. We also propose the Freeze Stethoscope, which selectively freezes parts of the model during fine-tuning. By controlling which modules are updated, we provide convincing empirical support for earlier insights and clues into the localization of reasoning within LLMs.

With different gadgets we propose in SfN, we provide not only sanity check level tests for our hypothesis, but also more convincing circumstantial supports and even direct empirical evidences. In short, the contributions in this paper are two-fold:

- With various diagnosis evidence (SfN), we are confident in hypothesizing that the output projection `o_proj` is mainly responsible for the reasoning in LLMs. The impact of this finding include not only potential ways to improve LLM that reasons (e.g., training much faster), but may generalize to produce better LLMs for other tasks (e.g., for a vertical LLM designed specifically for a domain). Our further conjecture is that other modules combined together lead to lucid conversations, but `o_proj` is less important in conversational ability.

- The proposed Stethoscope for Networks (SfN) gadgets are a set of tools that are useful in understanding modern LLMs and even other networks, which have the potential to enhance our understanding of LLM or deep neural network and may lead to alternative routes for further deep learning research.

## 2 Key Hypothesis: Output Projection is the Key for Reasoning

To present our findings, we start by introducing necessary background information and notations, while discussions on related work are deferred to Section 5.

Modern LLMs [42, 5, 29] mostly consist of many Transformer blocks. A Transformer [44] block is composed of a multi-head self-attention (MHSA) module and a multi-layer perceptron (MLP) module. Components in MHSA include various projections, such as those for computing Q, K and V, denoted as `q_proj`, `k_proj`, and `v_proj`, respectively. The output projection (`o_proj`) produces MHSA's output. Components in the MLP are mainly linear projections: up, down, and gate [16, 42, 5] projections, denoted as `up_proj`, `down_proj`, and `gate_proj`, respectively. The computation process is defined as:

$$x_{\texttt{attn}} = w_{\texttt{o}} \left[ \text{Softmax} \left( \frac{(w_{\texttt{q}}x)(w_{\texttt{k}}x)^{\top}}{\sqrt{d}} \right) (w_{\texttt{v}}x) \right]$$
$$x_{\texttt{mlp}} = w_{\texttt{down}} \left[ \sigma(w_{\texttt{gate}}x) \odot (w_{\texttt{up}}x) \right] \tag{1}$$

For simplicity, we omit residual connections and present the computation at the token level, without using matrix or vectorized notation. Other essential components not explicitly included in equation 1 include rotary positional embeddings (RoPE)[39], input embeddings (`embed_tokens`), layer normalization[4] (`layernorm`), and the language modeling head (`lm_head`).

Let $A$ be an LLM with weak or no reasoning ability. By carefully procuring a dataset of reasoning examples [13, 25, 52], one can cleanse and improve the quality of the dataset into the training data $\mathcal{D}$, and then finetune the existing model $A$ by using techniques such as SFT. The resulting LLM, model $B$, exhibits strong reasoning capabilities. For example, in commonly adopted practices, the base LLM $A$ is typically a widely used open-source model such as Qwen2.5-Math-1.5B, 7B or Qwen2.5-14B, 32B [53]. The reasoning model $B$ denotes a publicly available reasoning-enhanced variant, such as DeepSeek-R1-Distill-Qwen-1.5B, 7B, 14B, 32B [13], which comes with a clearly specified base model and well-documented training procedure. Models that are either not open-sourced [13, 40], or open-sourced without sufficient training details [41] or access to the base model [52], are not discussed in this paper.

### 2.1 The Delta Stethoscope

In the above scenario, it is obvious that $A$ and $B$ share exactly the same network architecture and structure, with their sole difference being the weights (parameters) inside various components. Suppose $w(A)$ ($w(B)$) denotes the set of weights for all modules in $A$ ($B$). Then, it is natural to conclude that to understand the difference between $A$ and $B$ (i.e., reasoning or not), we should focus on the difference between $w(A)$ and $w(B)$. Hence, we propose our first Stethoscope for Network.

**Assumption 1 (The Delta Stethoscope)** *Suppose $A$ and $B$ are two LLMs with weak and strong reasoning ability, respectively, and $B$ is obtained by finetuning from $A$. Then $w(B) - w(A)$ contains essential information if we want to pinpoint the source of the reasoning ability in $B$.*

For each component $X$ (e.g. $X = $ `q_proj`), we compute the $\ell_2$ norm of the weight difference, $\|w_X(B) - w_X(A)\|_{\ell_2}$, and visualize the results across all the blocks in Figure 2. For simplicity and due to space constraints, we present three representative comparisons: $A$ is Qwen2.5-Math-1.5B [54] or Qwen2.5-14B, 32B [53] and $B$ is DeepSeek-R1-Distill-Qwen-1.5B, 14B, 32B [13]. Additional results for other model sizes (7B and 8B) are provided in the appendix and exhibit similar patterns.

For the 1.5B models, the signal is less clear, but `o_proj` still exhibits a distinct pattern compared to `q,k,v_proj`—showing the largest change within the attention module and the second-largest across the entire model. As model size increases to 14B and 32B, this trend becomes more pronounced. In both cases, the most notable observation is that when $X = $ `o_proj`, the $\ell_2$ norm is at least two times larger than any other component, indicating the substantial changes in this module during reasoning enhancement.

In Figure 3, we further analyze the distribution of relative weight changes $\frac{w_X(B) - w_X(A)}{w_X(A)}$ for each linear module. To improve clarity and visual appeal, we plot the distribution every 5 layers and clip values in the range $[-1.0, 1.0]$ to mitigate the influence of outliers. The vertical axis represents the

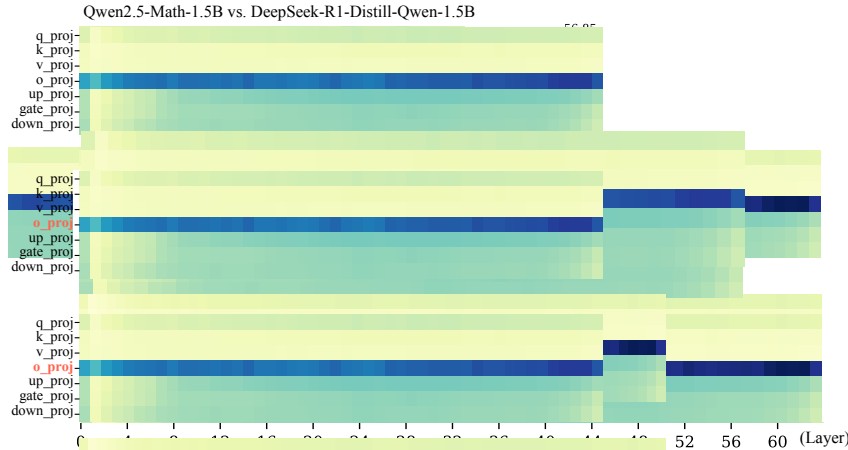

**Figure 2: Per-module L2 distance of linear weights between models** $A$ **and** $B$**.** Notably, the `o_proj` module shows the second-largest change in 1.5B models, and the largest in 14B and 32B models, highlighting its potential importance for reasoning. Similar trends are observed in 7B and 8B models (see appendix).

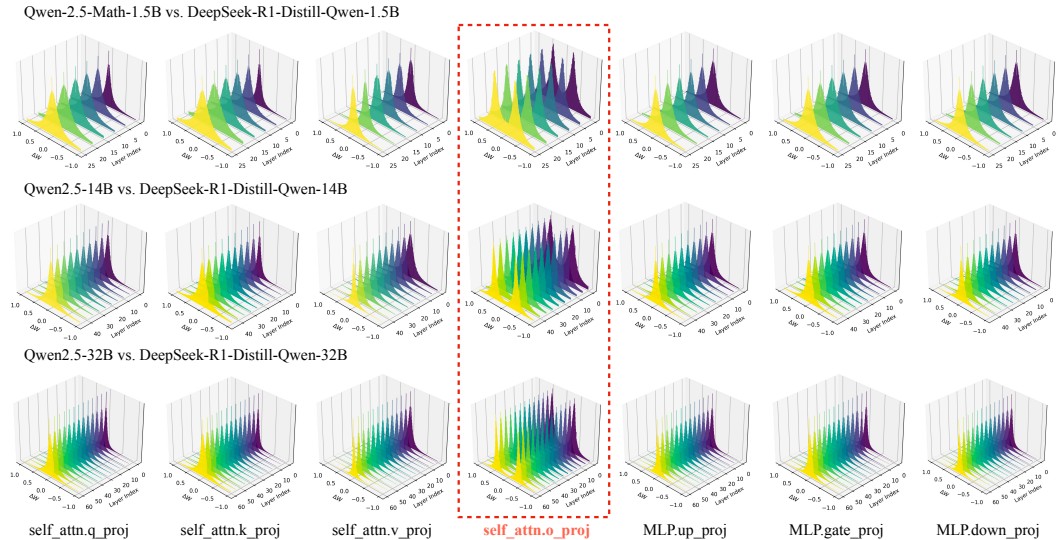

**Figure 3: Layer-wise distribution of relative weight changes between models** $A$ **and** $B$**.** While most modules display a unimodal distribution, the `o_proj` module uniquely exhibits a bimodal distribution, highlighting its distinctive behavior. Consistent patterns are observed across models of other sizes, with detailed results provided in the appendix.

frequency. A striking and consistent finding is that all linear modules—except `o_proj`—exhibit a unimodal distribution centered around zero, whereas *`o_proj` uniquely displays a clear bimodal pattern*, highlighting its distinct role.

Both observations hold consistently across model sizes and base models: `o_proj` exhibits the largest or second-largest weight shift, and the overall weight difference patterns remain strikingly similar. Therefore, it is reasonable to guess that the output projection `o_proj` plays a pivotal role in curating $B$'s reasoning ability. We are, however, not aware of `o_proj`'s specific role: is it solely responsible for reasoning? Or, is it collaborating with another module(s)? Or, in the worst scenario, is this difference in $\|w_X(B) - w_X(A)\|_{\ell_2}$ and $\frac{w_X(B) - w_X(A)}{w_X(A)}$ coincidental?

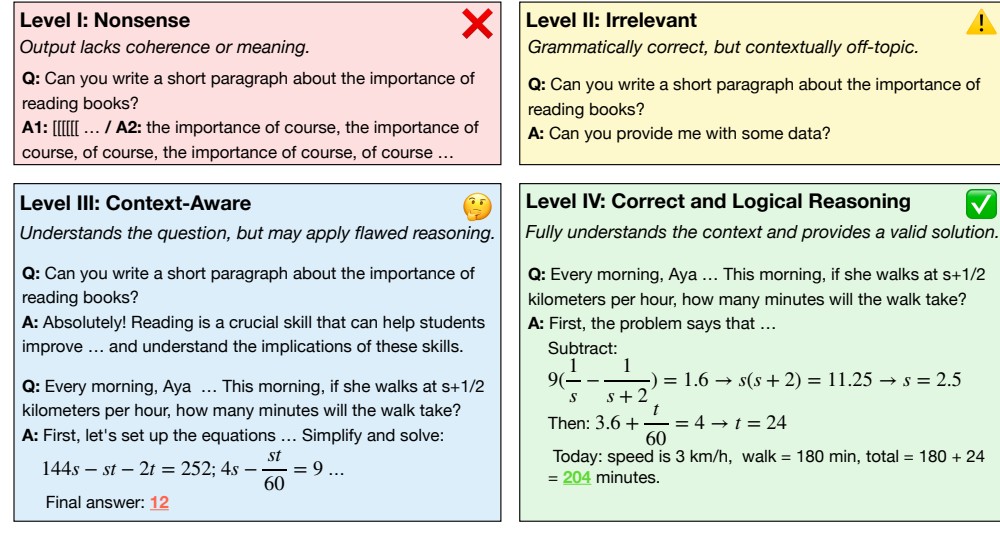

Figure 4: **Four levels of responses generated by the LLM**. From level I to level IV, the model exhibits stronger language organization and logical reasoning skills. Each example includes a question (e.g., a math problem from AIME or a typical user-issued request) and the corresponding response generated by the LLM.

## 2.2 The Merge Stethoscope

We design another gadget, the Merge Stethoscope, to answer this question. Suppose an LLM $M$ is formed by merging models $A$ and $B$, that is, $M$ has the same structure as $A$ and $B$, while a subset of its modules' parameters come from $A$ and the rest from $B$. In a conversational or reasoning task, what will the output of $M$ look like? We can imagine 4 levels of different output, as

Level I   A sequence of random or nonsense tokens.

Level II   A sequence that looks like normal sentences, but does not fit into the context of the task.

Level III   A sequence that is meaningful sentences that match the task's context well but will fail to reason in difficult problems.

Level IV   A sequence that reasons—and reasons correctly in most cases.

Figure 4 shows examples of level I to IV outputs. It is worth highlighting that $M$ is *rudely* merged from $A$ and $B$ *without any further tuning*. Hence, the intuitive conjecture will be that $M$ will produce level I output (i.e., ushering meaningless tokens). However, if model $M$, when merged in a specific configuration, is capable of producing level IV outputs for questions that model $A$ fails to solve, then the specially merged components are likely critical for reasoning.

**Assumption 2 (The Merge Stethoscope)** *Suppose $M$ is created by merging the output projection (`o_proj`) weights of $B$ (which has strong reasoning ability) and all other components of $A$ (which is weak in reasoning), and further suppose that $M$ has stronger reasoning ability compared to $A$. Then, we assume `o_proj` is crucial in achieving reasoning in LLMs.*

We attempt a minimal or atomic merge by replacing only the `o_proj` modules in model $A =$ Qwen2.5-Math-1.5B [54] with that of model $B =$ DeepSeek-R1-Distill-Qwen-1.5B [13], keeping all other components unchanged. Although we initially expected the resulting model to produce level I or level II outputs, the results turn out to be surprising. On the AIME 2024 benchmark [19], the merged model $M_1$ achieves level IV performance on several questions that model $A$ cannot solve. As shown in Table 1, the merged model not only yields correct reasoning and answers, but also tends to generate longer and more detailed responses compared to $A$. In contrast, replacing other modules such as {q,k,v}_proj and mlp leads to performance degradation. For example, model $M_2$, which replaces {q,k,v}_proj, produces level III outputs, while $M_3$, which replaces mlp, deteriorates to level I. Only replacing `o_proj` results in a correct reasoning process and a correct answer, as illustrated in Figure 5. This striking difference motivates our further investigation in Section 3.

| Model | Replaced Module | AIME 2024 | Average Tokens |
|---|---|---|---|
| $A$ (Q-1.5B) | - | 0.067 | 2421 |
| $M_1$ | `o_proj` | 0.200 | 5418 |
| $M_2$ | `{q,k,v}_proj` | 0.000 | 2058 |
| $M_3$ | `mlp` | 0.000 | 15532 |
| $B$ (D-1.5B) | - | 0.233 | 11892 |

Table 1: **AIME 2024 accuracy of the base model, the reasoning model, and their merged variants.** Each merged model is constructed by replacing specific modules in model $A$ with the corresponding module from model $B$.

**Q**: Every morning, Aya does a 9 kilometer walk … if she walks at s+1/2 kilometers per hour, how many minutes will the walk take?

$M_1$: To solve this problem, we need to determine … So, the walk will take 204 minutes, including the 24 minutes at the coffee shop. The final answer is **204**.

$M_2$: To solve this problem … output 12.0000000000000. The output indicates that the time taken for the walk is 12 minutes. So, the final answer is **12**.

$M_3$: … walking speeds increase speeds faster walking speeds increase walking speeds faster walking speeds faster walking …

Figure 5: **Examples of outputs generated by merged models.** Only $M_1$ produces both a valid reasoning process and the correct answer.

These results clearly show that the merged model $M$ has a stronger reasoning capacity than $A$, despite that $M$ is sutured from two completely different models and has *never* being finetuned. Now we feel confident in our assumption that `o_proj` is the key component responsible for reasoning in LLMs.

## 2.3 The Freeze Stethoscope

As models $A$ and $B$ scale up (e.g., to 7B parameters), merging components such as `q,k,v_proj` or `mlp` still results in significant performance degradation. However, unfortunately, merging `o_proj` no longer brings notable improvements in solving complex mathematical problems—although it does not harm accuracy, and still increases the generated output length.

Our analysis of $||w_X(B) - w_X(A)||_{\ell_2}$ suggests that this is due to a substantial mismatch in normalization parameters (that is, `layernorm` modules) between $A$ and $B$ at larger scales, compared to smaller models (e.g. 1.5B). Even when we merge both `o_proj` and `layernorm` parameters from $B$, the resulting model $M$ still fails to reason effectively, probably because the remaining parameters of $A$ are incompatible with the normalization parameters of $B$. To investigate this hypothesis in larger LLMs, we introduce the Freeze Stethoscope.

**Assumption 3 (The Freeze Stethoscope)** *Suppose that an LLM $F$ is obtained by supervised fine-tuning using the dataset $\mathcal{D}$. $F$ is initialized from $A$, and both `o_proj` and normalization components are tuned while other components are frozen. If $F$ exhibits strong reasoning ability, then we assume that `o_proj` is crucial in achieving reasoning in LLMs even in large-scale models.*

It is worth noting that `embed_tokens` and `lm_head` are also tuned.[2] Normalization module parameters are unfrozen by default. We adopt the pipeline of s1 [25] as our baseline, which uses the base model $A$ = Qwen2.5-32B-Instruct and the dataset $\mathcal{D}$ = s1K containing 1,000 high-quality reasoning traces. The results are shown in Table 2, where our model $F_4$ corresponds to model $B$ in Assumption 3. We do *not* strictly follow the training or testing setup of s1, primarily due to limited computational resources and the lack of an exact testing recipe to reproduce the reported results. However, our objective is not to optimize accuracy via testing tricks or prompt tuning, but to highlight the effectiveness of `o_proj` tuning compared to full-parameter tuning. For fair comparison, we adopt the "Budget Forcing Wait 2x" setting from s1 and retain all configurations without hyperparameter tuning.

Using this simplest possible experimental setup, Table 2 clearly shows that simply tuning `o_proj` and `layernorm` (model $F_2$)) leads to strong reasoning ability, while at the same time only tuning `layernorm` (model $F_1$) harms the reasoning of the LLM. Further unfreezing the parameters of `{q,k,v}_proj` (model $F_3$) yields little additional gain or even negative impact.

The training loss curves are shown in Figure 6. When all parameters including MLP are unfrozen, the model exhibits clear signs of overfitting, likely using the large MLP capacity to memorize the training set. In contrast, tuning only `o_proj` yields a smoother and more stable curve. Combined

---

[2]Without tuning these components, finetuning failed to converge.

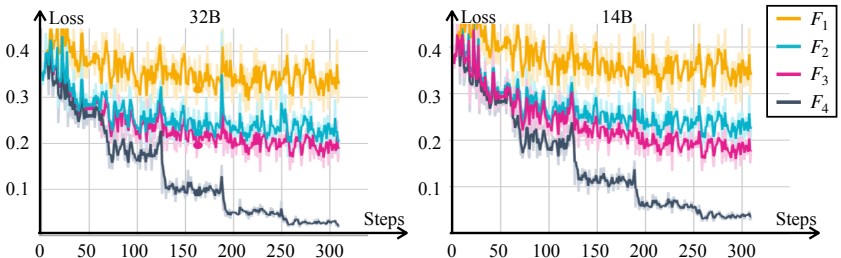

Figure 6: **Training loss curves for fine-tuning Qwen2.5-14B,32B-Instruct on reasoning tasks.** Different models unfreeze different sets of parameters, as detailed in Table 2.

| Model | Fintuned Modules | #Param (B) | Steps/s | AIME 2024 | Math 500 | GPQA Diamond |
|-------|------------------|-----------|---------|-----------|----------|--------------|
| $A$ (Q-32B) | - | - | - | 0.167 | 0.836 | 0.485 |
| $F_1$ | Emb + Head | 1.5 | 0.055 | 0.200 | 0.756 | 0.444 |
| $F_2$ | Emb + Head + o_proj | 3.2 | 0.052 | 0.367 | 0.890 | 0.520 |
| $F_3$ | Emb + Head + {q,k,v,o}_proj | 5.6 | 0.044 | 0.300 | 0.886 | 0.525 |
| $F_4$ ($B$) | All | 32.8 | 0.015 | 0.367 | 0.906 | 0.591 |
| $A$ (Q-14B) | - | - | - | 0.133 | 0.810 | 0.449 |
| $F_1$ | Emb + Head | 1.5 | 0.106 | 0.133 | 0.722 | 0.414 |
| $F_2$ | Emb + Head + o_proj | 2.8 | 0.099 | 0.266 | 0.848 | 0.485 |
| $F_3$ | Emb + Head + {q,k,v,o}_proj | 3.7 | 0.081 | 0.233 | 0.854 | 0.490 |
| $F_4$ ($B$) | All | 14.7 | 0.053 | 0.266 | 0.872 | 0.530 |

Table 2: **Reasoning performance of different fine-tuning strategies on Qwen2.5-{14B, 32B}-Instruct.** Emb denotes embed_tokens, Head denotes lm_head, and Attn denotes the entire MHSA. #Param refers to the number of trainable parameters, Steps/s indicates training speed, and the last three columns report commonly used metrics for evaluating reasoning models.

with its competitive performance, this suggests that the model learns to reason rather than simply memorize. Hence, we are now prepared and feel supported to propose our key hypothesis:

**Hypothesis 1 (Outstanding Output Projection)** *In an LLM that reasons well, we hypothesize that the output projection (o_proj) component is the single or at least the most important module that dominates its reasoning ability.*

With carefully chosen tuning strategy and hyperparameters, there is reason to believe that tuning only o_proj (+LN) can reach the level of model $B$ in terms of reasoning performance. And, beyond exhibiting reasoning abilities, Table 2 also shows that tuning only o_proj (+LN) has other significant advantages: e.g., significantly faster finetuning (3 times faster) and smaller GPU memory consumption. These advantages will become more established when larger LLMs are tuned.

## 3 Conjecture: Conversation Hinges on Other Modules but Not Output

We are mainly concerned with two abilities of LLMs: conversation and reasoning, which map to level III and IV in our categorization of LLM's outputs, respectively. Our Hypothesis 1 is on reasoning, but are there one module or several modules accounting for lucid conversations? In this section, we further propose a new stethoscope to diagnose this question and raise our conjectures accordingly.

### 3.1 The Destruction Stethoscope

Our previous stethoscopes follow a "constructive proof" style, while now we resort to the "proof by contradiction" style. If one module in an LLM is "destructed", and the LLM can still produce level III conversation outputs, then we have good reasons to guess that this module is not important in conversational ability; while it is important if the LLM ceases to dialogue regularly.

**Assumption 4 (The Destruction Stethoscope)** *Suppose a module $X$ is destructed (i.e., its normal functionality is disabled by some destruction method) in an LLM $A$. We denote the resulting LLM as*

| Destruction Method | Module | Output Level | Destruction Method | Module | Output Level |
|---|---|---|---|---|---|
| | q_proj | I | | q_proj | I |
| | k_proj | I | | k_proj | I |
| | v_proj | III | | v_proj | II |
| Zero | o_proj | III | ReInit | o_proj | III |
| | up_proj | I | | up_proj | I |
| | gate_proj | I | | gate_proj | I |
| | down_proj | I | | down_proj | I |
| Remove | – | I | | | |

Table 3: **Output levels of different modules under the three destruction methods: `Zero`, `ReInit`, and `Remove`.** All experiments are based on Qwen2.5-32B with destruction applied to specific layers.

$D$. Then, the fact that $D$ continues (or ceases to) produce level III output (meaningful sentences in the conversation's context) indicates whether $X$ is important for conversational abilities or not.

We propose 3 destructors to destroy a module:

`Zero` Set all parameters within $X$ to 0.

`ReInit` Re-initialize all parameters inside $X$ using Gaussian random numbers (mean=0, std=0.02).

`Remove` Remove the entire layer.

The `Zero` destructor is often equivalent to setting the output activation of $X$ to zeros (e.g., in a linear module like `o_proj`). We want to emphasize that `ReInit` incurs more serious damages to an LLM than `Zero` does. `Zero` may change activations to zero, but `ReInit` exerts random effects (i.e., noise) to LLM activations. What is more important, these random effects will act as input to the next Transformer block and the noise is quickly amplified. Hence, *level I or II output is expected* when $X$ is destroyed (especially when reinitialized) in a large number of Transformer blocks.

## 3.2 Conjectures Concerning the Conversation Capability

For model Qwen2.5-32B with 64 layers, we observe that destroying modules in early or late layers— where input and output representations are more sensitive—consistently yields level I outputs. To avoid this, we restrict destruction to blocks 5–30. This range is empirically chosen, as affecting more layers often causes all outputs to degrade to level I, making distinctions between modules impossible.

The experimental results are presented in Table 3. Specifically, we destroy selected modules and analyze the corresponding output. The `Remove` destructor removes the transformer layers as a whole. Note that the results are not statistics computed in many different experiments—it only reflects the conversation illustrated in Figure 4, but we observed similar patterns for other conversations.

Table 3 reveals distinct roles of modules in conversation. Notably, `o_proj`—crucial for reasoning— appears unimportant for conversation. In contrast, all MLP components (`up_proj`, `down_proj`, `gate_proj`) are essential. Within MHSA, `q_proj` and `k_proj` are important, while `v_proj` plays a minor role. Based on these (admittedly weaker) observations, we propose the following conjecture.

**Conjecture 1 (Division of Labor)** *Based on current observations, an LLM can be roughly divided as two sets of modules: output projection (`o_proj`) and all others, where `o_proj` is mainly responsible for reasoning and other modules for conversation.*

Then, output projection plays a unique role if this conjecture holds. Hence, we further propose another conjecture for it.

**Conjecture 2 (Output Projection Plugin)** *With conversational capabilities provided by other (frozen) modules, output projections may act as a plugin. For example, one set of `o_proj` for reasoning, and another set of `o_proj` for migrating an LLM to a vertical domain.*

# 4   Potential Implications and Applications

This paper mainly diagnoses LLMs from a theoretical, highly abstract perspective. However, our hypothesis and conjectures can also have highly practical implications and applications as long as they are correct or at least partially hold.

- **Fast and better reasoning LLMs**. By finetuning only `o_proj`, we can potentially find a better reasoning LLM with much faster training and much smaller GPU memory footprint.
- **Integrating non-reasoning and reasoning LLMs.** There is a recent trend to integrate chatting and reasoning LLMs into one model [52]. When we finetune a base LLM into a reasoning one using the previous procedure, they only differ in `o_proj`, `layernorm`, `embed_tokens` and `lmhead`, which occupy only 10% of model size. Hence, the two LLMs are easily loaded as one LLM with two sets of these module for different purposes.
- **Vertical LLMs**. Similarly, when equipped with different output projection plugins, one may adeptly obtain vertical LLMs for different domains.
- **Understanding deep neural networks.** The proposed Stethoscopes for Networks might be useful gadgets to understand other deep models, and new stethoscopes can be further developed. They will be potentially useful in diagnosing existing networks and even in providing alternative directions to future deep learning research.

# 5   Related Work

**Large Language Models.**   Modern LLMs such as GPT [29, 6], LLaMA [42, 43], Qwen [5, 53], and other representative models [7, 20] adopt an auto-regressive architecture and have demonstrated impressive capabilities across a wide range of natural language processing tasks, including question answering [32, 22], summarization [26, 27], and translation [51]. These models are typically trained on large-scale corpora using next-token prediction objectives, and their performance has been shown to scale with model size [21]. Further improvements in alignment and usability have been achieved through instruction tuning [28, 9, 47] and reinforcement learning from human feedback (RLHF) [8, 30], enabling more controllable and helpful dialogue generation.

**Reasoning Models.**   While LLMs exhibit emergent reasoning abilities [48, 35], recent efforts have further enhanced these capabilities through fine-tuning and architectural modifications [36, 56]. Chain-of-thought prompting [49] encourages intermediate reasoning steps, improving performance in arithmetic tasks, while self-consistency decoding [46] improves robustness by sampling multiple reasoning paths. Inspired by OpenAI's o1 [18], most advanced models now employ reinforcement learning [37, 30] to generate long reasoning traces with sparse rewards. This leads to significant improvements, particularly in complex math, code, and other professional domains [13, 52]. Despite these advances, the origin and location of reasoning ability in LLMs remain underexplored.

**Interpretability of LLMs.**   Understanding the inner workings of LLMs has attracted growing interest. Prior efforts include attention visualization [45], probing [15], and model editing [24, 34], with the aim of interpreting internal representations. Other studies decompose the behavior of the model into attribute functions to specific modules [11]. The "Physics of Language Models" series [1, 2, 3] investigates LLMs through controlled setups to reveal empirical and universal laws that dictate LLM behavior. However, these studies often exclude the most advanced models or focus on narrow, synthetic settings, offering limited insight into real-world models. Their findings provide little practical guidance for understanding reasoning in state-of-the-art models.

# 6   Conclusions

This work investigates a fundamental question in understanding large language models (LLMs): Is there a component or several components that are responsible for achieving the reasoning ability in LLMs? If the answer is affirmative, which components are responsible for the improvement?

We hypothesize that the output projection (`o_proj`) module plays a central role in enabling reasoning capabilities. To support this, we propose *Stethoscope for Networks (SfN)*, a diagnostic framework

that encompasses several probing techniques. Through the proposed `Delta`, `Merge`, `Freeze`, and `Destruction` stethoscopes, we observe consistent patterns indicating that `o_proj` is critical for reasoning, while other modules primarily support conversational fluency. These findings open new directions for efficient and modular LLM training.

Our findings are primarily based on a limited set of model families and reasoning benchmarks, and may not generalize to all architectures or tasks. Some diagnostic results rely on qualitative assessments rather than statistical validation. Furthermore, while the role of `o_proj` is empirically highlighted, a theoretical understanding of its function in reasoning remains to be established.

## Acknowledgments and Disclosure of Funding

This work was partly supported by the National Natural Science Foundation of China under Grant 62276123.

JW proposed the assumptions (Stethoscopes for Networks), hypothesis and conjectures. JS started this line of research in our group, proposed the `Zero` destructor, and effectively supported our main findings with experimental results. JW and JS wrote the paper.

We thank Ke Zhu for discussions.

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

# Technical Appendices and Supplementary Material

## A    Experimental Details

We primarily utilize open-sourced models to conduct experiments in this work. Given that DeepSeek-R1 is one of the most widely adopted reasoning models, and its authors have released a series of distilled models based on R1 [13], including both the specified base and finetuned reasoning models, we adopt their configurations in our study. Specifically, we use the DeepSeek-R1-Distill-Qwen [13] models with sizes of 1.5B, 7B, 14B, 32B and 70B as our reasoning models, and select Qwen2.5-Math-1.5B, 7B [54], LLaMA3.1-8B [12], Qwen2.5-14B, 32B [53] or Llama-3.3-70B-Instruct [12] as base models. All models are loaded and run using the Transformers library [50].

Our evaluation framework is based on the lm-evaluation-harness package [10]. To accelerate inference, we use vLLM [23] as the backend, which may slightly affect performance due to backend-specific optimizations. In the Merge Stethoscope experiments, we observe that the "chat" interface often generates irrelevant or nonsensical responses, while the "generate" interface produces coherent and contextually appropriate outputs. We suspect this discrepancy arises from misinterpreted system prompts. Therefore, we rely on the "generate" interface and implement a custom evaluation toolkit.

For the Freeze Stethoscope experiments, we build on the codebase of s1[25]. We use a learning rate of 1e-5, weight decay of 1e-4, a batch size of 16, and train for 5 epochs. Due to hardware limitations (i.e., lack of access to 16 H100 GPUs), we leverage DeepSpeed[33] with ZeRO Stage 3[31] to enable efficient training. The base model used here is Qwen2.5-32B-Instruct[53]. Evaluation is again conducted with lm-evaluation-harness, following the modified pipeline by the authors of s1, which disables generation of the end-of-thinking token and optionally appends the string "Wait" to the reasoning trace to encourage model reflection. We adopt the Budget Forcing "Wait" ×2 as our default testing configuration.

All visualization and inference experiments on 1.5B–14B models are conducted on a single NVIDIA A100 GPU. For training and evaluating 32B-70B models, we use a cluster of 8 NVIDIA A100 GPUs. Training typically takes around 6 hours, while testing on a single dataset usually requires about 2 hours.

## B    More Experimental Results

In the main paper, we present visualization results for the 1.5B, 14B, and 32B models. Here, we supplement those results by providing additional visualizations for the 7B, 8B, and 70B models. Following the Delta Stethoscope pipeline, we visualize both the absolute weight shift $|w_X(B) - w_X(A)|_{\ell_2}$ and the relative weight shift $\frac{w_X(B) - w_X(A)}{w_X(A)}$. The absolute weight shifts are shown in Figure 7, and the relative weight shifts are presented in Figure 8. The trends observed in the main paper remain consistent across these additional models. Notably, `o_proj` consistently exhibits the largest weight shift, with the effect being especially pronounced in the 70B model. Moreover, `o_proj` is the only module that displays a bimodal distribution in the relative weight shift.

## C    Statistical Significance and Broader Impacts

We report appropriate information regarding the statistical significance of our experiments. While we do not primarily focus on classical significance tests such as p-values, we provide multiple forms of empirical evidence—such as consistent module-specific weight shifts, response-level comparisons under controlled manipulations, and loss curves under different tuning strategies—that collectively establish the robustness of our findings. These analyses serve as a practical alternative to traditional error bars or confidence intervals and help substantiate our key claims.

This research has both promising benefits and important risks to consider. On the positive side, the proposed Stethoscope for Networks (SfN) framework provides a novel set of tools for interpreting LLMs, especially by localizing specific capabilities—such as reasoning—to individual components like the output projection (o_proj). These tools may significantly improve our understanding of LLMs, enabling more transparent, modular, and efficient model development. For instance, if reasoning

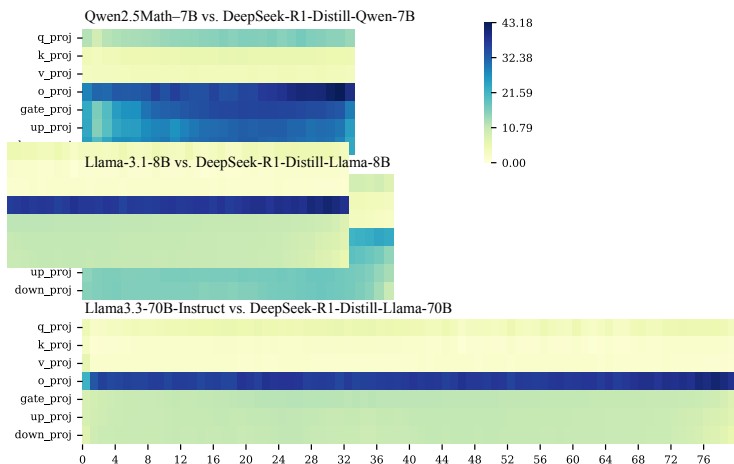

Figure 7: **Per-module L2 distance of linear weights between models** $A$ **and** $B$**.** Notably, the `o_proj` module shows the largest in 7B, 8B and 70B models, highlighting its potential importance for reasoning.

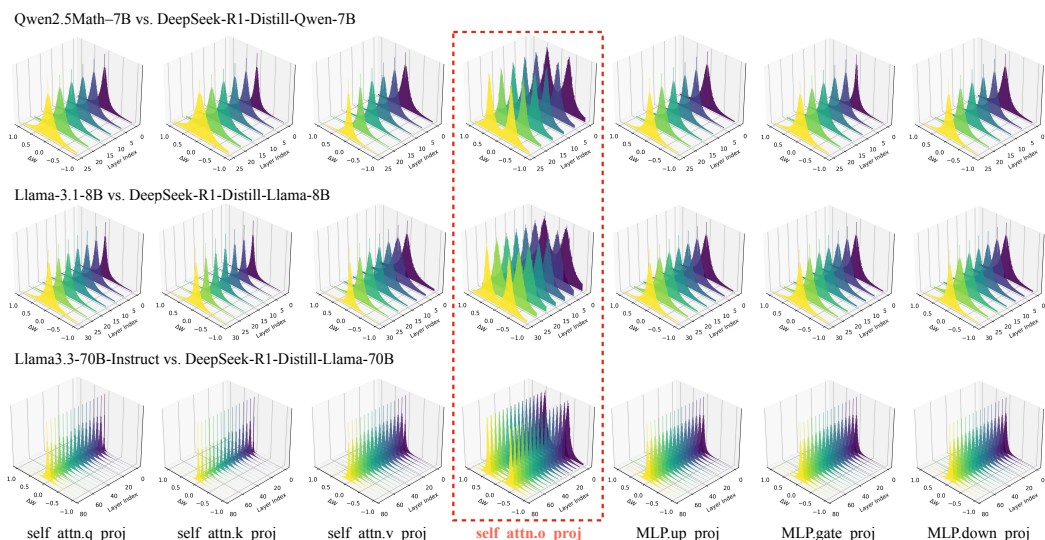

Figure 8: **Layer-wise distribution of relative weight changes between models** $A$ **and** $B$**.** While most modules display a unimodal distribution, the `o_proj` module uniquely exhibits a bimodal distribution, highlighting its distinctive behavior.

abilities can be enhanced by tuning a small subset of parameters, it could greatly reduce computational costs and increase accessibility for developing domain-specific or lightweight models.

However, this line of work also carries potential risks. Precisely identifying and isolating reasoning-related components might lower the barrier for targeted manipulation, such as unauthorized transfer or removal of reasoning abilities across models. This could facilitate misuse scenarios, including capability extraction, tampering, or model theft. Furthermore, while the diagnostic methods proposed aim to support interpretability, there is a risk that they may be overinterpreted, leading to an inflated sense of model transparency that does not generalize across architectures or tasks.

