# OpenReview forum: "Who Reasons in the Large Language Models?"
_NeurIPS.cc/2025/Conference — NeurIPS 2025 poster_

### Official Review · Reviewer_bgq1 · 2025-07-01

**Clarity:** 4
**Significance:** 3
**Originality:** 2
**Rating:** 5
**Confidence:** 4

**Summary:**

This paper investigates the reasoning of LLMs, namely, which part of the LLM is responsible for the reasoning abilities.
They do so by using 4 investigation tools given a base model A and a reasoning model B: comparing the weights of A and B, merge the weights of A and B, finetuning A with frozen components, destroying components in A.

They pinpoint the reasoning module to the output_projection of multi-head self attention.

**Questions:**

Did you try measuring the impact of the "output_proj" layer on tasks that do not involve reasoning?

**Ethical Concerns:**

["NO or VERY MINOR ethics concerns only"]

**Final Justification:**

I leave my score as it is a good paper worthy of being at NeurIPS

**Limitations:**

The only benchmarks used are reasoning ones, curious what is the effect on non-reasoning benchmarks

**Quality:**

4

**Strengths And Weaknesses:**

* Strength *

1. This is an interesting research question
2. It has large potential of influence on training of LLMs
3. Nice methodology

* Weaknesses *

My main remark is that the one of the roles of the output_projection in the MHSA layer is to map from the attention outputs to the mlp inputs. I'm not convinced that the lack of effect of the "q, k, v" in the merge setup is not due to their lack of importance but a mismatch between their outputs and mlp inputs.

---

> ### Author Rebuttal · Authors · 2025-07-30
>
> Thank you for your comments and suggestions.
>
> ---
> ### W1
>
> > the lack of effect of the "q, k, v" in the merge setup is … a mismatch between their outputs and mlp inputs.
> >
>
> Thank you for sharing your comments. Indeed, we have considered the possibility you raised. In the merge setup you mentioned, LayerNorm is not merged across all configurations. Therefore, the potential mismatch between the attention output and the MLP input exists not only when merging {q,k,v}_proj, but also when merging o_proj. **In both cases, the input to the MLP deviates from its original distribution, so the comparison remains fair**. Even under this mismatch, merging o_proj still yields substantial improvements, which underscores its unique importance.
>
> Moreover, the role of o_proj—as you pointed out—is to bridge attention outputs and MLP inputs, and this is precisely why we believe it plays a more crucial role in enabling reasoning compared with {q,k,v}_proj.
>
> ---
>
> ### Q1 & L1
>
> > … on tasks that do not involve reasoning?
> >
>
> > … what is the effect on non-reasoning benchmarks
> >
>
> We further extended our experiments using the freeze stethoscope method on several non-reasoing tasks to test the generality of our conclusions. The experimental setup is consistent with that outlined in the main paper. We present results under three configurations: the original pretrained baseline, o_proj-only fine-tuning, and full-parameter fine-tuning.
>
> In the domain of **chemistry**, we examine whether the observed importance of o_proj transfers to specialized scientific tasks. We fine-tune models on the SMolInstruct dataset [1] and evaluate performance on three benchmarks: I2F (converting IUPAC names to molecular formulas), I2S (converting IUPAC names to SMILES strings), and SIDER (predicting drug side effects). Mistral-7B is used as the base model. While we do not fully match the results reported in the original SMolInstruct paper, we take care to maintain consistency and fairness across all experimental settings. Fine-tuning only the o_proj layer yields strong performance while modifying far fewer parameters, highlighting its practical value in domain-specific applications.
>
> | Fintuned Modules | I2F | I2S | SIDER |
> | --- | --- | --- | --- |
> | - (Base model) | 0.000 | 0.000 | 0.381 |
> | o_proj | 0.616 | 0.518 | 0.598 |
> | All | 0.632 | 0.521 | 0.658 |
>
> In the field of **climate science**, we further assess the generalizability of our method using the SFT dataset and benchmark proposed in [2]. For this experiment, we use LLaMA-3.1-8B as the backbone model and evaluate its performance on two task dimensions: generating answers and interacting with tools. The findings indicate that fine-tuning solely the o_proj layer yields results on par with full-parameter tuning, once again underscoring the pivotal function of o_proj in adapting large models to specialized domains.
>
> | Fintuned Modules | Answer | Tool Usage |
> | --- | --- | --- |
> | - (Base model) | 0.391 | 0.483 |
> | o_proj | 0.787 | 0.563 |
> | All | 0.792 | 0.598 |
>
> I hope these experiments demonstrate the effectiveness of our method on non-reasoning tasks. **If further experiments or discussion are needed, we would be happy to continue the conversation!**
>
> ---
>
> [1] Yu, Botao, et al. "Llasmol: Advancing large language models for chemistry with a large-scale, comprehensive, high-quality instruction tuning dataset." arXiv preprint arXiv:2402.09391 (2024).
>
> [2] Lyu, Bohan, et al. "Adapting While Learning: Grounding LLMs for Scientific Problems with Tool Usage Adaptation." *Forty-second International Conference on Machine Learning*.

---

> > ### Comment · Reviewer_bgq1 · 2025-08-04
> > **Thank you for your answer**
> >
> > I thank you for your answer, I however remain unconvinced regarding the first part. Mainly, when you merge the o_proj as well, you can't be sure that it doesn't map the q,k,v projections to some universal to the mlp output. Namely, q,k,v,o are blocks that go together, with the o_proj fitting some specific q,k,v but mapping to some universal MLP distribution

---

> > > ### Author Response · Authors · 2025-08-05
> > > **Response to Reviewer bgq1**
> > >
> > > Thank you for your comments, which are useful for our future exploration.
> > >
> > > To be honest, we do not fully understand the role of q, k, v compared to that of o_proj. The results of our merging experiments do indicate that o_proj is very important, but they do not clarify the role of q, k, v.
> > >
> > > As we stated in Line 162, the merge stethoscope ceased to work in larger models (such as the 7B model). That is why we further proposed the freeze stethoscope.
> > >
> > > Hence, there are several possibilities: there might be a mechanism that allows o_proj to match the existing q, k, v and MLP; or there could be other underlying mechanisms. In this paper, we show that o_proj plays a crucial role, but as mentioned in our response to Reviewer GJy8, We only have hypotheses about why o_proj is so important, without solid experimental evidence to support them.
> > >
> > > We are very interested in exploring this point further. However, from a practical perspective, we believe our finding on the role of o_proj is already significant for research on reasoning LLMs, especially in various applications. From a theoretical perspective, while we currently lack a solid understanding of the mechanisms behind these empirical findings—and the broader community also has limited theoretical insight into LLMs—we are committed to working on this in the future.

---

> ### Author Response · Authors · 2025-08-09
> **Response to Reviewer bgq1 (update)**
>
> Thank you again for your reviews. we understand your concern regarding the merging experiments. While the merge stethoscope provides an initial intuition that o_proj plays an important role in reasoning, we believe that experiments using the freeze stethoscope offer more solid evidence. Here, we present **additional freeze stethoscope experiments** that may help address your concerns.
>
> As described in the main paper, we unfreeze specific modules and train on s1K. By examining each module’s improvement on reasoning benchmarks, we can infer its contribution to reasoning ability. embed_tokens and lm_head are unfrozen by default (following the same settings as in the paper). Compared with the experiments in the paper, we add another comparison—unfreezing {q, k, v}_proj only, without o_proj. We also conduct experiments on LLaMA-3.1-8B. The results are as follows.
>
> | Base Model | Fintuned Modules | AIME2024 | MATH 500 | GPQA Diamond |
> | --- | --- | --- | --- | --- |
> | Qwen2.5-14B | - | 0.133 | 0.810 | 0.449 |
> |  | o_proj | 0.267 | 0.848 | 0.485 |
> |  | {q,k,v}_proj  | 0.167 | 0.822 | 0.454 |
> |  | {q,k,v,o}_proj  | 0.233 | 0.854 | 0.490 |
> ||||||
> | Qwen2.5-32B | - | 0.167 | 0.836 | 0.485 |
> |  | o_proj | 0.367 | 0.890 | 0.520 |
> |  | {q,k,v}_proj  | 0.133 | 0.838 | 0.495 |
> |  | {q,k,v,o}_proj  | 0.300 | 0.886 | 0.525 |
> ||||||
> | LLaMA-3.1-8B | - | 0.067 | 0.456 | 0.283 |
> |  | o_proj | 0.133 | 0.584 | 0.323 |
> |  | {q,k,v}_proj  | 0.067 | 0.480 | 0.298 |
> |  | {q,k,v,o}_proj  | 0.133 | 0.588 | 0.333 |
>
> Across all models, **fine-tuning o_proj consistently outperforms fine-tuning {q, k, v}_proj**. o_proj is highly efficient in enhancing reasoning ability relative to its parameter size (results for full-parameter fine-tuning are provided in the main paper). These experiments indicate that o_proj plays a substantially more important role than {q, k, v}_proj in reasoning models, without needing to worry about the mismatch issue or the specific roles of {q, k, v, o}_proj.
>
> We hope this helps address your concern!

---

### Official Review · Reviewer_gHUg · 2025-07-03

**Clarity:** 1
**Significance:** 2
**Originality:** 2
**Rating:** 2
**Confidence:** 4

**Summary:**

The paper empirically studies the hypothesis that the model reasoning capabilities attribute to output projection’s parameters in the model multi-head self-attention. The proposed method, Stethoscope for Networks (SfN), compare the weight differences between fine-tuned and base models under controlled modules of merging, tuning, or destruction, and provides interpretable insights into the origin of capabilities like reasoning.

**Questions:**

1.  How did you unfreeze o_proj, without unfreezing q/k/v_proj? Did you just not update the gradients? If you are not changing what the model pays attention to but how the attention module can map the attended results as o_proj?
2. Table 2: F2 is based on attention layers updated on pre-trained model activations (by default), no? Did you mean you didn't update the proj_o with the final layer after updating with activations with premature layers? If so, isn't it naturally expected that F2 and F4 shows small changes; F4 achieves a bit of performance gains as it is shown?
3. What do authors think their novel contributions are from this work, that are beyond the previous observations (MLP=factual storage, attention=reasoning logic, layer updates=all parts contribute, even if frozen at inference)?

**Ethical Concerns:**

["NO or VERY MINOR ethics concerns only"]

**Final Justification:**

The author response was somewhat disappointing.

While the tone was mostly professional, it included dismissive remarks (e.g., “I don’t know have you heard PEFT or LoRA”) and an inappropriate appeal to other reviewers’ scores to suggest my rating was a misunderstanding. More importantly, when clarification was provided in good faith, the authors did not acknowledge or engage with it, even though they responded to later follow-ups from other reviewers.

Accordingly, several key concerns were deflected, particularly the misinterpretation of post-hoc weight changes in o_proj as causal evidence for reasoning localization. Instead of addressing the conceptual flaws, the response largely reiterated procedural details.

Given these concerns and after reviewing all the rebuttals, I lean more towards rejection to acceptance.

**Limitations:**

Layer updates mean "reasoning" isn't isolated. Their conclusion risks a localization fallacy that tuning o_proj improves reasoning. But the experimental setups/results are a bit unclear/weak to conclude that o_proj is the place the reasoning happens. It still seems that they use frozen model activations (which model stores their factual associations) and observing weight changes or successful performance after tuning o_proj with those factual associations do not seem sufficient evidence that reasoning is localized here but rather convincing to say it is a whole computation of model circuits.

**Quality:**

2

**Strengths And Weaknesses:**

### **Strengths**

1. Interesting idea under "Stethoscope for Networks (SfN) metaphor. Delta/Merge/Freeze/Destruction works well with the ideation with parameter isolation.
2. Instead of existing work with factual recall, syntax, token, prediction, or attention layers, the work tries to narrow down their scope of exploration even further down into components of attention modules (o_proj, q/k/v_proj).
3. The question is valuable to ask where the reasoning is localized/attributed primarily.

### **Weaknesses**
1. Writing needs more work/ esp. introduction (motivation) is a bit chaotic and not compelling.
- LLMs still struggle with trial-and-error experiments for developing new capabilities -> can you provide more examples? what challenges are we aiming to tackle?
- Black-box style success -> can we be explicit how it is problematic to recent successes in reasoning capabilities of PPO, DPO, SFT, etc?
2. The experiments are not riguorous; not compelling if the research question is addressed enough. Table 2: F2 is based on attention layers updated on pre-trained model activations (by default) but without being updated with the final layer. This is naturally expected that F2 and F4 shows small changes.
3. The originality of the work is not quite strong. Recent works not limited to classic works [a, b] show that MLP often act like "fact retriever" or local updaters while attention moves information around, enabling reasoning chains and variable binding. But that doesn't mean the reasoning only happens in attention outputs as the attention layer updates with model activations that are frozen yet pre-trained knowledge. Not sure how this contribute novel findings beyond already existing works.


ref
[a] Transformer Feed-Forward Layers Are Key-Value Memories, EMNLP 2021
[b] Locating and Editing Factual Associations in GPT, NeurIPS 2022

---

> ### Author Rebuttal · Authors · 2025-07-31
>
> ### W1
>
> > Writing ...
> >
> > - LLMs … trial-and-error experiments … -> … provide more examples …
> > - Black-box … -> … how it is problematic …
>
> These comments are not intended to dismiss the remarkable achievements of LLMs, but rather to highlight that there remains considerable room for improvement. I have hands-on experience pre-training and fine-tuning LLMs at the 8B scale. My typical training pipeline involves collecting large volumes of high-quality data, followed by empirical validation through training runs. However, in practice, the outcome is often **unpredictable**—some data that appear to be high quality fail to yield meaningful improvements on specific benchmarks. Based on this experience, I believe the development of LLMs still involves substantial trial and error.
>
> Indeed, SFT, RLHF, and reinforcement learning for reasoning have all demonstrated significant success. However, we still lack a deep understanding of **how** these models actually work. As Geoffrey Hinton has repeatedly emphasized, LLMs remain largely opaque—we still know very little about their internal mechanisms [1]. The **debate over reasoning capabilities** further underscores this opacity. For example, works such as [2,3,4] argue that reasoning success can be attributed to data contamination or memorization rather than true reasoning. However, many researchers strongly disagree, and the debate is ongoing.
>
> Regardless of which side is correct, the core issue is clear: **we lack a mechanistic understanding of LLMs, especially for reasoning tasks.** We believe treating them as black boxes is insufficient. Our work aims to shed light on the inner workings of reasoning in LLMs through novel analytical techniques.
>
> As for the writing, we believe the logic and structure of the paper are clear and coherent. Notably, the other three reviewers rated the paper’s **clarity as “3: good,” “4: excellent,” and “4: excellent.”** In contrast, your rating of “1: poor” suggests there may have been a misunderstanding during your review. **If further clarification is needed, we would be more than happy to engage in discussion.**
>
> ---
>
> ### W2 & Q1 & Q2
>
> > Table 2: F2 is based on attention layers updated on pre-trained model activations (by default) but without being updated with the final layer. This is naturally expected that F2 and F4 shows small changes.
> >
>
> > Table 2: F2 is based on attention layers updated on pre-trained model activations (by default), no? Did you mean you didn't update the proj_o with the final layer after updating with activations with premature layers? If so, isn't it naturally expected that F2 and F4 shows small changes; F4 achieves a bit of performance gains as it is shown?
> >
>
> We apologize, but **we are genuinely unclear about the exact meaning of your questions.** Nevertheless, we have attempted to interpret them to the best of our understanding. **If our interpretation does not align with your intention, we would be happy to engage in further clarification.**
>
> First, when you mention that “F2 is … updated on pre-trained model activations without being updated with the final layer” could you please clarify what you mean? As far as we understand, this seems to reflect a misunderstanding of the actual computation process. Our SFT is performed based on the model’s **final output**. The training procedure is **identical to** **standard** **SFT** for LLMs or typical end-to-end training in neural networks
>
> It’s unclear whether you mean that F2 shows only small changes compared to F4, or that both F2 and F4 exhibit small changes compared to the baseline A. If you meant the former, it is not self-evident that F2 and F4 would yield only minor differences. For example, in the 32B model, F2 involves just **3.2B** unfrozen parameters, while F4 has **32.8B**—**ten times more**. If you meant the latter, that also doesn’t seem accurate, as Table 2 clearly shows that both F2 and F4 achieve **significant improvements on reasoning benchmarks** compared to the baseline A.
>
> > How did you unfreeze o_proj, without unfreezing q/k/v_proj? Did you just not update the gradients? If you are not changing what the model pays attention to but how the attention module can map the attended results as o_proj?
> >
>
> **Unfreeze only a part of parameters is very common** in LLMs and diffusion. (I don’t know have you heard PEFT or LoRA). In our setting, we unfreeze only the o_proj parameters while keeping the {q,k,v}_proj frozen. This is done by setting `requires_grad=True` for `o_proj` and `requires_grad=False` for the others.
>
> What exactly do you mean by “if you are not changing what the model pays attention to but how the attention module can map the attended results as o_proj”?  To clarify, let’s write out the attention computation:
>
> $$
> x_{\text{attn}} = \text{LayerNorm} \left( w_o \left[ \text{Softmax} \left( \frac{(w_q x)(w_k x)^\top}{\sqrt{d}} \right) (w_v x) \right] +x\right)
> $$
>
> $$
> x_{\text{mlp}} = w_{\text{down}} \left[ \sigma(w_{\text{gate}} x) \odot (w_{\text{up}} x) \right] + x
> $$
>
> In our paper, the o_proj refers to the **weight matrix** $w_o$, which is the **linear layer** following the attention mechanism. Even when only o_proj is unfrozen, **the attention maps can still be indirectly affected.** For example, updating the o_proj in the first layer modifies its output, which in turn alters the input to the next layer—including its attention map. Thus, even if only o_proj is unfrozen, its influence can still propagate **throughout the model.**
>
> ---
>
> ### W3 & Q3
>
> > originality ...  classic works [a, b] … But that doesn't mean the reasoning only happens in attention outputs …
> >
>
> > … novel contributions … that are beyond the previous observations …
> >
>
> We would like to emphasize that the roles of various modules in LLMs—including even the division of responsibilities between attention and MLP layers—remain open research questions that are still actively studied in the community [5,6]. Moreover, while the works you referenced are indeed insightful, they were published **prior to the emergence of the first dedicated reasoning models** (such as o1). As such, their research objectives differ fundamentally from ours. These earlier studies were not designed to address the central question we focus on: **who or what is responsible for reasoning** *in reasoning-capable LLMs*.
>
> In contrast, our work explicitly addresses this question. If you search for it online, you will find that this question **has not been clearly posed or investigated before**. We believe this is a novel and important research direction, especially given the current ongoing debates around whether reasoning in LLMs is real or merely a byproduct of memorization or data leakage.
>
> Additionally, the hypothesis you mentioned—such as "attention equals reasoning logic"—has not been supported by concrete experimental validation, to our knowledge. Our work is the **first** to systematically pose this question and provide a suite of tools to explore it. Our findings may contribute to more efficient training or the integration of thinking and non-thinking models. We believe it is a meaningful and timely contribution to the field.
>
> ---
>
> ### L1
>
> > … the experimental setups/results are a bit unclear/weak … they use frozen model activations … do not seem sufficient evidence ...
> >
>
> First, we would like to clarify that **activations are not frozen** in our experiments—we believe there may have been a fundamental misunderstanding of our methodology throughout your review.
>
> Second, we have conducted a **comprehensive series of experiments** across a wide range of models. Specifically, we performed **delta stethoscope** analyses on the following pairs:
>
> - Qwen2.5-Math-1.5B vs. DeepSeek-R1-Distill-Qwen-1.5B
> - Qwen2.5-Math-7B vs. DeepSeek-R1-Distill-Qwen-7B
> - LLaMA-3.1-8B vs. DeepSeek-R1-Distill-LLaMA-8B
> - Qwen2.5-14B vs. DeepSeek-R1-Distill-Qwen-14B
> - Qwen2.5-32B vs. DeepSeek-R1-Distill-Qwen-32B
> - LLaMA-3.3-70B-Instruct vs. DeepSeek-R1-Distill-LLaMA-70B
>
> We also extended our analysis to **Qwen3-235B-A22B-Instruct-2507** and **Qwen3-235B-A22B-Thinking-2507** (see our response to Reviewer 6DXm), and consistently found that **o_proj shows the most significant changes** across all these model comparisons.
>
> In addition to delta stethoscope, we applied the **merge stethoscope** and **destruction stethoscope** methods. For the **freeze stethoscope**, we conducted experiments on **Qwen2.5-14B** and **Qwen2.5-32B** with reasoning tasks, and further extended our evaluation to **LLaMA-3.1-8B**, **Mathstral-7B**, and **Mistral-7B** across **reasoning, chemistry, and climate** domains.
>
> Across all experiments, the results consistently support our conclusion that **o_proj is the most critical component for enabling reasoning and other vertical (domain-specific) capabilities**. You mentioned that our evidence is insufficient—we genuinely welcome the opportunity to provide **additional experiments or analyses** to clarify or strengthen our conclusions. Please let us know if there are particular aspects you’d like us to elaborate on.
>
> ---
> [1] https://www.newyorker.com/magazine/2023/11/20/geoffrey-hinton-profile-ai
>
> [2] Shojaee, Parshin, et al. "The illusion of thinking: Understanding the strengths and limitations of reasoning models via the lens of problem complexity." *arXiv preprint arXiv:2506.06941* (2025).
>
> [3] Shao, Rulin, et al. "Spurious rewards: Rethinking training signals in rlvr." *arXiv preprint arXiv:2506.10947* (2025).
>
> [4] Wu, Mingqi, et al. "Reasoning or Memorization? Unreliable Results of Reinforcement Learning Due to Data Contamination." *arXiv preprint arXiv:2507.10532* (2025).
>
> [5] Allen-Zhu, Zeyuan, and Yuanzhi Li. "Physics of language models: Part 3.1, knowledge storage and extraction." *arXiv preprint arXiv:2309.14316* (2023).
>
> [6] Dong, Yihe, et al. "Attention Retrieves, MLP Memorizes: Disentangling Trainable Components in the Transformer." *arXiv preprint arXiv:2506.01115* (2025).

---

> ### Comment · Reviewer_gHUg · 2025-08-04
>
> Thanks for the detailed response. I do believe the initial evaluation is valid. I will try to be more detailed on my review below; hope that helps.
>
> **W1 - writing clarity**; Writing is indeed not solid; esp. the current intro is wordy and lacks a clear, direct motivation. It instead cycles through vague statements without defining key terms.
>
> E.g.,
> - Lines 16–20: mention LLM reasoning capabilities and trial-and-error limitations.
> - Line 21: raise a “black-box style” issue without specifying what is black-box in this context.
> - Lines 22–28: cite RLHF/DPO/SFT models as examples of reasoning, without explaining why these methods are grouped or how their reasoning patterns relate.
> - Lines 28–30: repeat “underlying mechanism unexplained” without clarifying if your work aims to explain why reasoning emerges, how it works in RL/SFT, or both.
>   - additional question: did authors explain this research question in the paper? (why and how reasoning abilities? both in RL and SFT? why did you bring these two aspects; do they show (dis-)similar patterns?)
>
> The repeated vague phrasing (“black-box style,” “reasoning abilities,” “valuable to guiding future directions”) makes it difficult for readers to see the specific gap your work addresses. There are a lot of rooms required to improve the clarity of the writing. That way, it would be nicely connected to Cases 1-3.
>
> Next page shows the similar pattern of writing.
> - "...the above questions will be extremely valuable to guiding the future direction of LLM research. ... " like how? Generally your work uses vague claims and loose logics to support your ideas.
>
> - lines43-46 - "To support our hypothesis, we propose a few techniques for diagnosing LLM’s behaviors, in particular, the potential functionalities and impacts of various modules in it. We call these techniques Stethoscope ..." -> What are "a few techniques for diagnosing LLM’s behaviors"? The authors referred readers to Figure 1 but no properly specified textual explanations in the text and the figure. In the caption of the figure, it says that the framework is to identify which components of an LLM give rise to specific abilities" but it became more ambiguous than reasoning capabilities in the earlier introduction.
>
> Overall, wording and explanations are not specific enough and missing focal points for readers to find it informative.
>
> Since you mentioned other reviewers' high scores on clarity, I must admit that I was a bit ***surprised*** about their high scores despite the current writing quality.
>
>
> **W2 & Q1 & Q2**
> I am not saying the technical availability. My main question is the core reasoining behind the experimental setups and results to support your work "localization of reasoinng".
>
> What your work is doing are:
> - The attention pattern (softmax over QK) and the values V come entirely from the pretrained model
> - Finetuning only o_proj cannot change what is attended to, because it can only linearly remap the already computed attention outputs (as you had expressed in your math).
> - any improvments in reasoning in F2 is due to remapping fixed upstream activations, not creating new reasoning computations.
>
> Why the weakneses still remain to support the authors' statement are:
> - The paper interpret the performance gains from o_proj tuning as localizing reasoning there but in F2 that's not logically valid because the reasoning signal still originates from frozen q/k/v computations.
> - In other words, tuning W_o cannot change what is attended, only remap the fixed attended outputs (as the authors admitted in your paper and the response here " Even when only o_proj is unfrozen, the attention maps can still be indirectly affected.")
> - Concluding localization of reasoning in W_o from F2 does not sound solid. Things that doesn't sound right from the paper: “we hypothesize that it is the output projection’s parameters (o_proj)… that is in charge of reasoning in an LLM”
>
> In conclusion, the claim that reasoning is "localized" in O_proj is ***unspported*** because, in F2, W_q, W_k, W_v remain frozen, so attention patterns and values come entirely from the pretrained model, meaning ***any gains come from remapping fixed upstream computation rather than creating or localizing new reasoning ability***.
>
> **L1**
> I get your experiments right -- please read all the sentence "observing weight changes or successful performance after tuning o_proj with those factual associations ..." was the key point. I understand your experiments, but observing weight changes or successful performance after tuning O_proj on top of frozen factual associations from the pretrained model is a weak setup for supporting a claim about reasoning localization, as it cannot distinguish between adapting a readout and actually computing reasoning there.
>
>
>
> I appreciate your hard work but still think the overall ratings are a bit higher than it should be. Happy to know if there are any misunderstandings.

---

### Official Review · Reviewer_6DXm · 2025-07-03

**Clarity:** 4
**Significance:** 3
**Originality:** 3
**Rating:** 4
**Confidence:** 3

**Summary:**

This paper investigates the underlying mechanisms of reasoning abilities in LLMs. The paper hypothesizes that the output projection module within the transformer multi-head self-attention is primarily responsible for reasoning capabilities. To support this hypothesis, authors introduce Stethoscope for Networks (SfN), a suite of diagnostic tools, including the Delta, Merge, Freeze, and Destruction Stethoscopes. Through empirical evidence from these tools, they suggest that the output projection module plays a central role in reasoning, while other modules contribute more to fluent dialogue. The findings offer insights into LLM interpretability and propose avenues for more targeted and efficient reasoning training strategies.

**Questions:**

- While the paper empirically demonstrates the importance of o_proj in reasoning, a deeper theoretical understanding of why o_proj plays this role is missing. Could the authors elaborate on potential mechanisms or computations within o_proj that might facilitate reasoning? For instance, does it act as a bottleneck for abstract representation, or does it aggregate information in a specific way that enables logical steps?
- The experiments are primarily conducted on Qwen2.5 models. While the results are consistent within this family, it would be beneficial to discuss the potential generalizability of the findings to other LLM architectures. Are there architectural differences that might influence the role of o_proj in reasoning, or do the authors believe this finding is broadly applicable across modern LLMs?
- The Freeze Stethoscope section notes that for larger models (e.g., 7B), simply merging o_proj is insufficient, and an o_proj + layernorm merge is considered. Could the authors expand on the specific interplay between o_proj and layernorm in enabling reasoning, especially for larger models? Understanding this interaction might reveal a more nuanced picture of reasoning localization.
- The paper mentions that fine-tuning with reasoning examples leads to strong reasoning capabilities in model B. Is there any hypothesis or preliminary observation on how the type of training data might influence the specialization of modules, particularly o_proj?

**Ethical Concerns:**

["NO or VERY MINOR ethics concerns only"]

**Final Justification:**

Thanks for the detailed responses, which addressed most of my concerns.

**Limitations:**

yes

**Quality:**

3

**Strengths And Weaknesses:**

The paper presents a novel hypothesis regarding the localization of reasoning ability in LLMs, specifically pointing to the o_proj module. The introduction of the Stethoscope for Networks (SfN) framework with its various diagnostic tools provides a systematic and empirical approach to probing LLM internals. They provided extensive experiments, and the consistency of observations across different model sizes adds to their findings. The paper is well-structured and generally easy to follow. The introduction clearly lays out the problem and the paper's hypothesis. Each Stethoscope method is explained with its underlying assumption and how it contributes to the overall hypothesis. The core hypothesis, attributing reasoning capabilities primarily to the o_proj module, is novel and deviates from common assumptions that reasoning stems from the entire model or more complex interactions. However, some recent and relevant studies seem to be missed in related work and discussion, for example:

```
Ruis, Laura, Maximilian Mozes, Juhan Bae, Siddhartha Rao Kamalakara, Dwarak Talupuru, Acyr Locatelli, Robert Kirk, Tim Rocktäschel, Edward Grefenstette, and Max Bartolo. "Procedural knowledge in pretraining drives reasoning in large language models." arXiv preprint arXiv:2411.12580 (2024).
```

---

> ### Author Rebuttal · Authors · 2025-07-30
>
> Thank you for your comments and suggestions.
>
> ---
>
> ### W1
>
> > … some recent and relevant studies …
> >
>
> Thank you. This paper offers valuable insights about reasoning models, and we will discuss it in the related work part.
>
> ---
>
> ### Q1
>
> > … potential mechanisms or computations within o_proj that might facilitate reasoning …
> >
>
> According to [1], the attention mechanism is chiefly responsible for gathering relevant contextual cues, while the MLP predominantly handles the storage and activation of internalized knowledge.
>
> Thus, we **guess** that the mechanism by which o_proj facilitates reasoning stems from its **strategic position** within the Transformer block—immediately following the attention computation and directly preceding the MLP. At this junction, o_proj receives inputs that contain rich, integrated cross-token information. Its outputs are then passed into the MLP, potentially activating internal knowledge representations.
>
> In this sense, o_proj acts as a **bottleneck within the block**—regulating **how much aggregated information** flows from the attention module into the MLP.
>
> We, however, did not include this guess in our paper, intentionally—it is difficult to find supportive evidences either mathematically or empirically for this guess. Before presenting it to the community, we want to convince ourselves first that this guess have a good chance to be true. We are not there yet.
>
> ---
>
> ### Q2
>
> > … generalizability of the findings to other LLM architectures.
> >
>
> We fully acknowledge the importance of demonstrating the generality and robustness of our findings. Beyond the core experiments based primarily on the Qwen series presented in the main text, we provide additional evidence using the delta stethoscope method on two model pairs: **LLaMA-3.1-8B vs. DeepSeek-R1-Distill-LLaMA-8B** and **LLaMA-3.3-70B-Instruct vs. DeepSeek-R1-Distill-LLaMA-70B**. As shown in Figures 1 and 2 of the **supplementary materials**, both comparisons consistently validate the conclusion that the **o_proj layer undergoes the most substantial changes.** In particular, this effect is most striking in LLaMA-3.3-70B-Instruct, where the distinction is significantly more pronounced than in other models.
>
> We further investigate this phenomenon in a MoE architecture by comparing two recent SOTA open-source models: **Qwen3-235B-A22B-Instruc-2507** and **Qwen3-235B-A22B-Thinking-2507**. We are excited to share our recent findings based on these two newly released models. These models differ significantly from standard dense architectures. Each consists of 94 layers, with 388 linear modules per layer. To compare the "instruct" and "thinking" variants, we compute the L2 norm of the weight differences for each type of linear module across all layers. Our analysis shows that the **o_proj module consistently exhibits the largest or second-largest change in every layer, most often ranking as the largest**. This consistently prominent shift in o_proj further supports its central role in enabling reasoning within Transformer models.
>
> We also conduct extended evaluations using the freeze stethoscope method across diverse model families. The setup aligns with that described in the main paper. We compare performance across three configurations: baseline, fine-tuning only o_proj, and full-parameter fine-tuning. Specifically, we applied this framework to **LLaMA-3.1-8B and Mathstral -7B**, and fine-tuned them on the s1k dataset using the same SFT protocol as in the main experiments. As reported in the table, fine-tuning solely the o_proj layer consistently delivers competitive performance, despite involving substantially fewer trainable parameters. This finding again highlights the central role of o_proj in enabling reasoning capabilities in large language models.
>
> To enhance the robustness of our findings, we conduct extended evaluations using the freeze stethoscope across diverse model families. The setup follows that described in the main paper. We compare three configurations: baseline, fine-tuning only the o_proj layer, and full-parameter fine-tuning, while keeping the embedding and lm_head layers unfrozen in all cases. Specifically, we apply this framework to **LLaMA-3.1-8B and Mathstral-7B**, fine-tuning them on the SFT s1K dataset using the same protocol as in the main experiments. As shown in the table, fine-tuning only the o_proj layer consistently yields competitive performance, despite involving significantly fewer trainable parameters. This result further underscores the central role of o_proj in supporting reasoning capabilities in LLMs.
>
> | Base Model | Fintuned Modules | AIME2024 | MATH 500 | GPQA Diamond |
> | --- | --- | --- | --- | --- |
> | **LLaMA-3.1-8B** | - | 0.067 | 0.456 | 0.283 |
> |  | o_proj | 0.133 | 0.584 | 0.323 |
> |  | All  | 0.133 | 0.592 | 0.354 |
> | **Mathstral  7B** | - | 0.067 | 0.566 | 0.126 |
> |  | o_proj | 0.100 | 0.614 | 0.166 |
> |  | All  | 0.133 | 0.602 | 0.157 |
>
> In summary, we have evaluated our findings across a range of models, including **LLaMA-3.1-8B** and **Mathstral -7B** (small models), **LLaMA-3.3-70B-Instruct** (a large dense model), and **Qwen3-235B-A22B-Instruct-2507** (a large-scale MoE model). These results collectively demonstrate that our conclusions are broadly applicable across modern LLM architectures.
>
> ---
>
> ### Q3
>
> > … specific interplay between o_proj and layernorm in enabling reasoning
> >
>
> That’s a good question. We consider the case of LayerNorm now. Can LayerNorm alone induce reasoning ability? The answer is **no**. Through both the merge stethoscope and freeze stethoscope analyses, we observe that LayerNorm itself does not directly contribute to reasoning. However, it still plays a vital **supporting role**. In most architecture of modern LLMs, the positions of o_proj and LayerNorm are as follows:
>
> $$
> x_{\text{attn}} = \text{LayerNorm} \left( w_o \left[ \text{Softmax} \left( \frac{(w_q x)(w_k x)^\top}{\sqrt{d}} \right) (w_v x) \right] +x\right)
> $$
>
> $$
> x_{\text{mlp}} = w_{\text{down}} \left[ \sigma(w_{\text{gate}} x) \odot (w_{\text{up}} x) \right] + x
> $$
>
> The LayerNorm sits directly **between o_proj and the MLP**. While it contains only a small number of parameters and therefore has limited capacity, it plays a **crucial role as a translator**—mapping the output of o_proj into a form that is compatible and interpretable by the downstream MLP.
>
> This translation becomes especially important in both the merge stethoscope and freeze stethoscope settings. In these setups, the MLP is held fixed while o_proj is updated. Such updates inevitably introduce a distribution shift in the output of o_proj, which can misalign with the expectations of the fixed MLP. To bridge this gap, it is essential to also update the LayerNorm. By doing so, we ensure that the transformed output remains aligned with the fixed MLP, allowing it to process the information effectively and **enabling reasoning** to emerge.
>
> ---
> ### Q4
>
> > … how the type of training data might influence the specialization of modules …
> >
>
> That’s exactly an aspect we are deeply interested in. First, it is important to highlight that **model size significantly influences module specialization**. As shown in Figure 2 of the main paper and Figure 1 of the supplementary material, we observe a clear pattern: when the model is smaller, the MLP layers exhibit larger parameter changes, while o_proj updates are relatively minor. This suggests that **smaller models, which lack sufficient capacity and internal knowledge, tend to rely more heavily on the MLP to memorize training data directly**.
>
> Assuming the model is **sufficiently large**, we then explore how the type of training data affects the functional specialization of different modules. Using Qwen2.5-32B as our base model, we fine-tune on diverse data types, including complex mathematical problems like **s1K**, domain-specific datasets such as **SMolInstruct** (chemistry) [2] and **climate science** data [3]. Across these experiments, we find that o_proj consistently shows the most significant updates, while MLP layers change only modestly. This suggests that when the model already possesses relevant knowledge, and the fine-tuning task involves **applying that knowledge to solve problems**, the o_proj layer becomes the primary site of adaptation.
>
> In contrast, when the training data contains entirely **new information** **that the model has never encountered before**, the MLP appears to take on a more dominant role. To investigate this, we follow the setup in [4] and fine-tune the model on the **bioR** dataset (a reproduction version on GitHub), which includes synthetic biographical data for 100,000 fictional individuals. Since this data introduces new knowledge rather than tasks requiring reasoning over existing concepts, we observe that the MLP layers undergo substantial updates, while attention-related modules remain relatively unchanged.
>
> Based on these observations, we have the following guess: when the task involves solving problems using existing knowledge, the o_proj layer is the most critical and undergoes the greatest changes. However, when the training data introduces novel knowledge, the model relies more heavily on the MLP to encode and internalize that information.
>
> ---
> [1] Dong, Yihe, et al. "Attention Retrieves, MLP Memorizes: Disentangling Trainable Components in the Transformer." *arXiv preprint arXiv:2506.01115* (2025).
>
> [2] Yu, Botao, et al. "Llasmol: Advancing large language models for chemistry with a large-scale, comprehensive, high-quality instruction tuning dataset." arXiv preprint arXiv:2402.09391 (2024).
>
> [3] Lyu, Bohan, et al. "Adapting While Learning: Grounding LLMs for Scientific Problems with Tool Usage Adaptation." *Forty-second International Conference on Machine Learning*.
>
> [4] Allen-Zhu, Zeyuan, and Yuanzhi Li. "Physics of language models: Part 3.1, knowledge storage and extraction." *arXiv preprint arXiv:2309.14316* (2023).

---

> > ### Comment · Reviewer_6DXm · 2025-08-07
> >
> > Thanks for the detailed responses, which addressed most of my concerns. I will maintain my original positive score.

---

> > > ### Author Response · Authors · 2025-08-07
> > > **Response to Reviewer 6DXm**
> > >
> > > Thank you for your review and comments. We sincerely appreciate your suggestions.

---

### Official Review · Reviewer_GJy8 · 2025-07-07

**Clarity:** 3
**Significance:** 3
**Originality:** 2
**Rating:** 4
**Confidence:** 4

**Summary:**

Summary: This paper investigates which components within large language models (LLMs) are primarily responsible for the emergence of reasoning abilities, particularly in mathematical and logical tasks. The authors propose a systematic diagnostic toolkit, "Stethoscope for Networks (SfN)," which includes several probing techniques to analyze the internal behavior of LLMs. Through comprehensive empirical studies, the paper shows that the output projection module (o_proj) in the Transformer’s multi-head self-attention mechanism plays a central role in reasoning, while other modules contribute mainly to conversational fluency. The findings present new insights into modularity and targeted training in modern LLMs.

**Questions:**

1、The analysis shows o_proj is key for reasoning in the tested models, but have you tried the same methods on models with different architectures (e.g., LLaMA, GPT-style, or encoder-decoder models) to see if this conclusion still holds?
2、In Section 3.2 (Destruction Stethoscope), the conversational ability analysis is mostly based on selected examples. Could you provide more systematic statistics or human evaluations to support the proposed "division of labor" between modules?
3、The paper mentions potential applicability of SfN to other networks (Section 4), but there are no experiments beyond reasoning tasks. Do you have preliminary results or plans to test SfN on abilities such as knowledge retention or code generation?

**Ethical Concerns:**

["NO or VERY MINOR ethics concerns only"]

**Final Justification:**

I decide to keep my score.

**Limitations:**

yes

**Quality:**

3

**Strengths And Weaknesses:**

Strengths:
1、The study introduces a simple yet effective methodology for attributing functional capabilities to specific modules inside LLMs.
2、The empirical analysis is thorough and well-supported by clear ablation and merge experiments.
3、Identifying o_proj as critical for reasoning could lead to more efficient and targeted fine-tuning approaches.
4、The proposed SfN toolkit may be broadly useful for interpretability research beyond reasoning tasks.
Weakness
1、Lack of theoretical explanation: The paper mainly relies on empirical analysis and does not provide a deeper theoretical understanding of why the o_proj module dominates reasoning in Transformers.
2、Limited generalization: Experiments are only conducted on a few open-source models such as Qwen and DeepSeek, and mainly on specific reasoning tasks (e.g., math problems), so it is unclear whether the findings generalize to other architectures and tasks.
3、Some conclusions rely on qualitative judgments: The division of modules for "conversational ability" is mainly based on output examples and observation, lacking large-scale statistical validation and being somewhat subjective.
4、Generality of the SfN tool is not fully validated: Although the authors propose SfN as a universal diagnostic toolkit, experiments are focused on reasoning modules and tasks, lacking evidence of its effectiveness on other types of models and tasks.

---

> ### Author Rebuttal · Authors · 2025-07-30
>
> Thank you for your valuable comments and suggestions.
>
> ---
> ### W1
>
> > … theoretical explanation …
> >
>
> We fully agree with you that theoretical explanation is missing, which is intentional though. As you mentioned in the review, our empirical analysis of the special role of o_proj is “well-supported”, but we really do not have a corresponding theoretical analysis yet. And, to be honest, theory for LLM in a whole is definitely much weaker than empirical results, which is not limited to our work.
>
> Furthermore, we have a guess (not even a hypothesis) on this. [1] suggests that attention is primarily responsible for retrieving information, while the MLP is responsible for memorization: attention operates across tokens and MLP functions within individual tokens.
>
> The unique importance of the o_proj layer in reasoning stems from its pivotal position within the Transformer block—**immediately following the attention computation and directly preceding the MLP**. Thus, it is possible (based on [1]) that o_proj receives inputs containing integrated cross-token information, which supports analytical capabilities, and its outputs feed into the MLP, potentially activating internal knowledge.
>
> This guess might be useful for understanding the theoretical side of o_proj, but we did not include it in our paper intentionally—it is difficult to find supportive evidences either mathematically or empirically for this guess. As such, we prefer to stick to empirical studies. a choice that seems to suit the current stage of LLM research.
>
> ---
> ### W2 & W4 & Q1 & Q3
>
> > … whether the findings generalize to other architectures and tasks.
> >
>
> > … lacking evidence of its effectiveness on other types of models and tasks.
> >
>
> > …  tried the same methods on models with different architectures … to see if this conclusion still holds?
> >
>
> > …  there are no experiments beyond reasoning tasks. …  test SfN on abilities such as knowledge retention or code generation?
> >
>
> Thank you for your suggestions. We agree that it is important to demonstrate the generalizability of our findings. In addition to the results presented in the main paper—primarily based on the Qwen and DeepSeek model series—we also include further evidence using the delta stethoscope method on both **LLaMA-3.1-8B vs. DeepSeek-R1-Distill-LLaMA-8B** and **LLaMA-3.3-70B-Instruct vs. DeepSeek-R1-Distill-LLaMA-70B**. These results, shown in **Figure 1** and **Figure 2** of the **supplementary material** along with our paper, consistently support the same conclusion as in the main text: the o_proj layer exhibits the most significant change. Notably, for LLaMA-3.3-70B-Instruct, this phenomenon is particularly clear and prominent compared to all other models.
>
> Furthermore, we are excited to present delta stethoscope results on **Qwen3-235B-A22B-Instruct-2507 vs. Qwen3-235B-A22B-Thinking-2507**, one of the most advanced open-source models, released just a few days ago. These models adopt a MoE architecture, which differs substantially from standard dense models. Each model consists of 94 layers, with each layer containing 388 linear modules. To compare the "instruct" and "thinking" variants, we compute the L2 norm of the weight differences for each type of linear module across all 94 layers. Our analysis reveals that the **o_proj module consistently exhibits the largest or second-largest change in every single layer—most often ranking as the largest.** This consistently significant shift in o_proj further reinforces our conclusion about its central role in supporting reasoning within Transformer models.
>
> In addition, we conducted more experiments using the **freeze stethoscope** method across different models and tasks to strengthen our findings. The experimental settings follow those described in the main paper. Specifically, we report results for the following configurations: baseline, o_proj fine-tuning and full-parameter fine-tuning. As in the original setup, both the embedding and lm_head layers are kept unfrozen by default.
>
> - **Reasoning on other architecture**: We further conduct experiments on reasoning tasks using an additional model. Specifically, we adopt **LLaMA-3.1-8B** as the baseline and perform SFT on the s1k dataset, following the same setup as described in the main paper. The results are shown in the table below. It is evident that fine-tuning only the o_proj layer achieves competitive performance with the significantly fewer trainable parameters, highlighting its critical role in reasoning tasks.
>
>
>     | Fintuned Modules | AIME2024 | MATH 500 | GPQA Diamond |
>     | --- | --- | --- | --- |
>     | - (Base model) | 0.067 | 0.456 | 0.283 |
>     | o_proj | 0.133 | 0.584 | 0.323 |
>     | All  | 0.133 | 0.592 | 0.354 |
>
> - **Chemistry:** We also investigate whether our approach generalizes to domain-specific capabilities (**vertical LLM models**). To this end, we use the SMolInstruct dataset [2] for SFT and evaluate the models on several molecular science tasks. We select three representative benchmarks: I2F (IUPAC name to molecular formula), I2S (IUPAC name to SMILES), and SIDER (side effect classification of drugs). The baseline model is **Mistral-7B** [3]. Although we do not fully reproduce the performance reported in the original work—possibly due to differences in training recipes or the total number of training steps—we ensure that all training settings and comparisons are fair and consistent. Notably, our o_proj fine-tuning approach still achieves competitive results.
>
>   | Fintuned Modules | I2F | I2S | SIDER |
>   | --- | --- | --- | --- |
>   | - (Base model) | 0.000 | 0.000 | 0.381 |
>   | o_proj | 0.616 | 0.518 | 0.598 |
>    All | 0.632 | 0.521 | 0.658 |
>
> - **Climate:** We further evaluate our method in another specialized domain—climate science—using the SFT dataset and benchmark introduced in [4]. In this experiment, we adopt **LLaMA‑3.1‑8B** as the base model and evaluate performance on two aspects: answer generation and tool usage. The results show that fine-tuning only the o_proj layer achieves performance comparable to full-parameter fine-tuning, highlighting the critical role of o_proj in adapting to domain-specific tasks again.
>
>
>     | Fintuned Modules | Answer | Tool Usage |
>     | --- | --- | --- |
>     | - (Base model) | 0.391 | 0.483 |
>     | o_proj | 0.787 | 0.563 |
>     | All | 0.792 | 0.598 |
>
> We are sorry that we do not have the resources (time, GPU, etc.) to finish investigation on code generation. But thank you for your suggestion—we do plan to finish it in the near future.
>
> ---
> ### W3 & Q2
>
> > The division of modules … lacking large-scale statistical validation
> >
>
> > In Section 3.2 (Destruction Stethoscope) … provide more systematic statistics or human evaluations
> >
>
> That’s a great suggestion—Thank you! To quantitatively assess the model’s capability in daily conversation, we evaluate its **perplexity (PPL)** on the **DailyDialog** dataset [5], which consists of multi-turn dialogues from everyday scenarios. Specifically, we use the test set to compute the PPL of a LLM.
>
> Following the destruction stethoscope experimental setup, we select Qwen2.5-32B as the baseline model and re-initialize specific modules within blocks 5 to 30. The resulting PPLs are reported in the table below, where the first row represents the model without any modifications.
>
> From the perspective of perplexity, we can directly observe the functional importance of each module in daily dialogue. Notably, the {q_proj, k_proj} and MLP modules are highly critical for conversational coherence, with MLP exerting the most significant impact. In contrast, v_proj has a relatively minor effect—its re-initialization still allows sentence generation, albeit with reduced logical consistency. Interestingly, **o_proj appears to have the least influence** in this context: **re-initializing it results in only slight degradation**, and the model can still produce largely coherent and natural responses. These experiments provide more reliable statistical evidence and further validate the conclusions drawn from the destruction stethoscope analysis.
>
> | Destructed Module | PPL | Destructed Module | PPL |
> | --- | --- | --- | --- |
> | - (Base model) | 10.8 |  |  |
> | q_proj | 212.9 | up_proj | 411.8 |
> | k_proj | 268.6 | gate_proj  | 232.4 |
> | v_proj | 97.2 | down_proj | 505.1 |
> | o_proj | 34.6 |  |  |
> ---
>
> [1] Dong, Yihe, et al. "Attention Retrieves, MLP Memorizes: Disentangling Trainable Components in the Transformer." *arXiv preprint arXiv:2506.01115* (2025).
>
> [2] Yu, Botao, et al. "Llasmol: Advancing large language models for chemistry with a large-scale, comprehensive, high-quality instruction tuning dataset." arXiv preprint arXiv:2402.09391 (2024).
>
> [3] Jiang, Albert Qiaochu et al. “Mistral 7B.” arXiv preprint arXiv:2310.06825 (2023).
>
> [4] Lyu, Bohan, et al. "Adapting While Learning: Grounding LLMs for Scientific Problems with Tool Usage Adaptation." *Forty-second International Conference on Machine Learning*.
>
> [5] Li, Yanran, et al. "Dailydialog: A manually labelled multi-turn dialogue dataset." *arXiv preprint arXiv:1710.03957* (2017).

---

> > ### Comment · Reviewer_GJy8 · 2025-08-06
> >
> > I have reviewed the author's response, which addresses most of my concerns. I will maintain my original positive score.

---

> > > ### Author Response · Authors · 2025-08-06
> > > **Response to reviewer GJy8**
> > >
> > > Thank you for your valuable feedback and suggestions. Your comments on W3 & Q2 is useful to us. We genuinely appreciate your review of our paper and rebuttal.

---

> > ### Comment · Reviewer_GJy8 · 2025-08-06
> >
> > Thanks for the detailed response, which resolved most of my concerns, so I decide to keep my positive score.

---

### Comment · Area_Chair_ENYp · 2025-08-05
**Rebuttal posted, please engage in discussion!**

Hi Reviewers,

The author has posted their rebuttal. Could you please review their response and share your thoughts?

Please engage in the discussion as soon as possible to allow for a meaningful back-and-forth before the deadline.

Thank you for your timely attention to this.

Best,

Your AC

---

### Note · Authors · 2025-08-12

Dear Reviewers and AC,

We are grateful for all your comments.

- Our research is considered “interesting”, “has large potential of influence” (Reviewer bgq1), "present new insights" (Reviewer GJy8), “novel” (Reviewer 6DXm).
- Our method is described as “simple yet effective, and may be broadly useful for interpretability research” (Reviewer GJy8), “systematic and empirical” (Reviewer 6DXm), “nice” (Reviewer bgq1).
- Our experiments are considered “thorough and well-supported” (Reviewer GJy8), “extensive” (Reviewer 6DXm).
- We do not agree with the comments from Reviewer gHUg, but appreciate the recognition such as “valuable question” and “interesting idea”.

Two concerns are the need for additional experimental support and theoretical analysis.

- Experimentally, we substantially extended core parts of our paper by incorporating extra models (Qwen3-235B-A22B-Thinking-2507, LLaMA-3.1-8B, Mistral-7B, and Mathstral 7B), settings, and tasks (reasoning tasks, vertical domains such as chemistry and climate, and general tasks such as dialogue and biography).
- On the theoretical side, we acknowledge that we currently have only a guess regarding why o_proj plays such an important role, likely due to its pivotal position within the transformer block. Developing a solid theoretical explanation is challenging and remains a topic for future research.
- We **thank the reviewers for your unique insights**. For example, “systematic statistics to support division of labor” (Reviewer GJy8), “specific interplay between o_proj and layernorm” (Reviewer 6DXm), and “effect of q, k, v in the merge setup” (Reviewer bgq1). We have addressed these questions individually.
- Reviewer gHUg raised issues such as “the intermediate attention maps and activations being entirely from the pretrained model”. We believe they are clearly inconsistent with our experiments, and have provided detailed explanations.

Overall, we believe our paper offers a new perspective and valuable insights into reasoning LLMs. Our method SfN is general and useful to analyzing and probing LLMs, and our experiments are extensive and multi-faceted. We believe our work can inspire further exploration in areas such as model merging (especially between reasoning and non-reasoning models), efficient finetuning, and interpretability research for LLMs.

We thank all the reviewers and the AC again for your feedback, which have made our work more solid and comprehensive.

Best regards,

Authors of submission 7822

---

### Decision · Program_Chairs · 2025-09-17

**Decision:**

Accept (poster)

**Comment:**

This paper presents a novel investigation into the localization of reasoning abilities within LLMs. The authors hypothesize that mathematical reasoning capabilities are not diffusely distributed throughout the model but are primarily concentrated in the output projection module (o_proj) of the Transformer's multi-head self-attention mechanism. To probe this hypothesis, they introduce a new suite of diagnostic tools named Stethoscope for Networks (SfN). Using SfN, the authors provide empirical evidence suggesting that the o_proj module is indeed central to reasoning tasks, while other components may be more involved in maintaining fluent and coherent dialogue.

The paper received a mixed set of reviews. After a careful reading of the paper and the reviewers' comments, I find myself agreeing with the more positive assessments. The work's primary contribution is its fresh perspective on LLM interpretability. The hypothesis that a complex capability like reasoning could be localized to a specific submodule is a bold and interesting one. While the evidence is indeed empirical, this is characteristic of much of the current research in deep learning, and framing the central claim as a "hypothesis" or "conjecture" is appropriate.

The findings, even though not theoretically proven, are likely to be of interest to the community and could inspire new lines of inquiry into the functional specialization of LLM components. Furthermore, the introduction of the SfN is a potentially valuable contribution in its own right. Diagnostic tools that allow for a more granular analysis of LLM internals are crucial for advancing our understanding. I believe for a conference like NeurIPS, there is value in accepting work that is thought-provoking and opens up new avenues for exploration, even if it is primarily empirical. This paper has the potential to spark important discussions and shift how we think about training and interpreting these complex models. Therefore, I lean towards acceptance.